# SDPGO: Efficient Self-Distillation Training Meets Proximal Gradient Optimization

**Tongtong Su[1], Liao Yun[2]\*, Fengbo Zheng[1]\***

[1]School of Computer and Information Engineering, Tianjin Normal University
[2]College of Artificial Intelligence, Tianjin University of Science and Technology

## Abstract

Self-knowledge distillation (SKD) enables single-model training by distilling knowledge from the model's own output, eliminating the need for a separate teacher network required in conventional distillation methods. However, current SKD methods focus mainly on replicating common features in the student model, neglecting the extraction of key features that significantly enhance student learning. Inspired by this, we devise a self-knowledge distillation framework entitled *Self-Distillation training via Proximal Gradient Optimization or SDPGO*, which utilizes gradient information to identify and assign greater weight to features that significantly impact classification performance, enabling the network to learn the most relevant features during training. Specifically, the proposed framework refines the gradient information into a dynamically changing weighting factor to evaluate the distillation knowledge via the dynamic weight adjustment scheme. Meanwhile, we devise the sequential iterative learning module to dynamically optimize knowledge transfer by leveraging historical predictions and real-time gradients, stabilizing training through mini-batch-based KL divergence refinement while adaptively prioritizing task-critical features for efficient self-distillation. Comprehensive experiments on image classification, object detection, and semantic segmentation demonstrate that our method consistently surpasses recent state-of-the-art knowledge distillation techniques. Code is available at: https://github.com/nanxiaotong/SDGPO.

## 1 Introduction

In the past few years, deep neural networks (DNNs) have achieved great success in computer vision tasks, including image classification [15, 45], object detection [22, 41], semantic segmentation [11, 38] and others. Due to the over-parameterization and high computational complexity of DNNs, deployment costs are increasingly significant. Knowledge distillation (KD) [14] addresses this by transferring knowledge from a large teacher model to a lightweight student via soft target alignment, uniquely enabling efficient compression and fast inference without structural modifications.

Despite its benefits, pre-training a high-capacity teacher network requires considerable computational sources and run-in memory. Furthermore, the capacity gap issue [49, 65, 26]has persistently hindered the development of knowledge distillation. In conventional offline KD, a one-way knowledge transfer is applied through a two-stage training process to steer the learning of the student network [21, 43, 12]. In contrast, online KD operates without dependence on a pre-trained teacher, instead enabling an ensemble of students to learn collaboratively in an end-to-end fashion [59, 28, 46, 52]. Consequently, existing KD methods, whether online or offline, tend to be computationally expensive and time-consuming, thereby hindering their deployment on resource-constrained end devices such as mobile phones and digital cameras.

---

\*Corresponding author: Liao Yun (yliao@tust.edu.cn), Fengbo Zheng (fzh229@tjnu.edu.cn)

39th Conference on Neural Information Processing Systems (NeurIPS 2025).

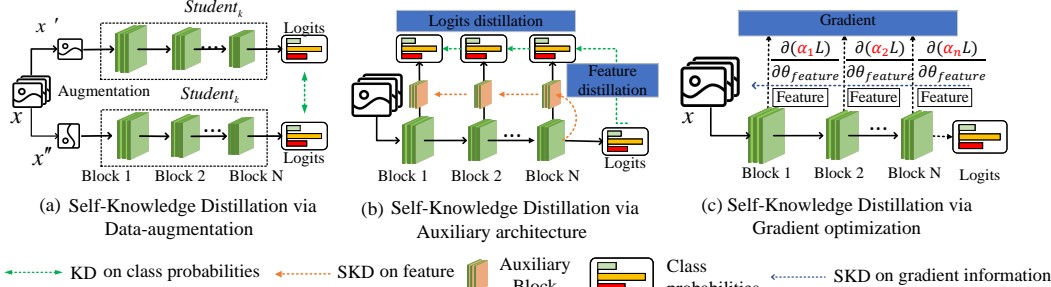

(a) Self-Knowledge Distillation via Data-augmentation

(b) Self-Knowledge Distillation via Auxiliary architecture

(c) Self-Knowledge Distillation via Gradient optimization

Figure 1: Brief comparison of our SDPGO with data-augmentation and auxiliary-architecture methods. SDPGO utilizes the gradients to identify and distill the most valuable and relevant knowledge.

To address these limitations, SKD [57] addresses these limitations by enabling self-distillation through a single network that acts as both a teacher and a student. As depicted in Figure 1(a), a prevalent framework in SKD employs auxiliary techniques to enhance the training signal. Augmentation strategies [48, 18] enhance model consistency by enforcing prediction invariance across different augmented views of the same input. Parallelly, historical predictions [36, 20] generated during earlier training iterations act as dynamic soft labels to regularize current updates. Another line of work introduces auxiliary network components [61, 54], where knowledge from deeper layers is distilled into shallower ones via dedicated pathways in Figure 1(b). However, existing SKD approaches often *rely on fixed or heuristically defined weights to prioritize features* during knowledge transfer. These weights remain static throughout the training process, thus failing to adapt to the evolving importance of features as the model learns.

In this work, we are committed to continuously evaluate feature importance through gradient magnitudes within each mini-batch, enabling instantaneous prioritization of task-critical features. Unlike static distillation methods, our method performs real-time refinement of the knowledge, as illustrated in Figure 1(c), enabling dynamic adaptation to evolving data distributions and training dynamics. Specifically, our method leverages real-time gradient analysis to dynamically shift its attention. During high-gradient phases, it amplifies the distillation loss for features that drive immediate performance gains. Conversely, in low-gradient phases, it recalibrates attention toward under-optimized features to prevent premature convergence.

Motivated by our findings, we propose Self-Distillation training via Proximal Gradient Optimization (SDPGO), a novel self-knowledge distillation framework that integrates sequential iterative learning with gradient-driven feature analysis. Our method uses a proximal weight assignment module to assess the importance of intermediate features via their gradient magnitudes from mini-batches. This mechanism, rooted in proximal gradient theory, enables real-time prioritization of high-impact features while adaptively suppressing less critical ones through sparsity-aware optimization. Meanwhile, the framework leverages sequential iterative updates to refine knowledge across training steps. Each iteration's KL divergence between current and previous predictions (from the latter half of mini-batches) creates a feedback loop, allowing the model to incrementally correct errors and consolidate accurate knowledge without additional computational overhead.

**Contributions.** This work overcomes existing SKD limitations in the utilization of dynamic critical features during training. In doing so, we make the following contributions:

- **Enables Real-Time Knowledge Refinement.** We propose a simple but efficient self-knowledge distillation based on proximal gradient optimization scheme, named as SDPGO, which leverages the gradients of each mini-batch to dynamically weigh the importance of features (see Figure 2). Unlike static SKD methods, SDPGO refines knowledge transfer in real time, eliminating reliance on external teachers and enabling the model to learn critical features directly from its own training trajectory.

- **Dynamic Gradient-Driven Weight Assignment:** We introduce a dynamic weight assignment scheme for self-distillation that dynamically evaluates the importance of extracted features by leveraging gradient magnitudes within each mini-batch. This scheme prioritizes features with larger gradients, thereby guiding the student model to focus on the most impactful knowledge and adapt to ongoing training dynamics.

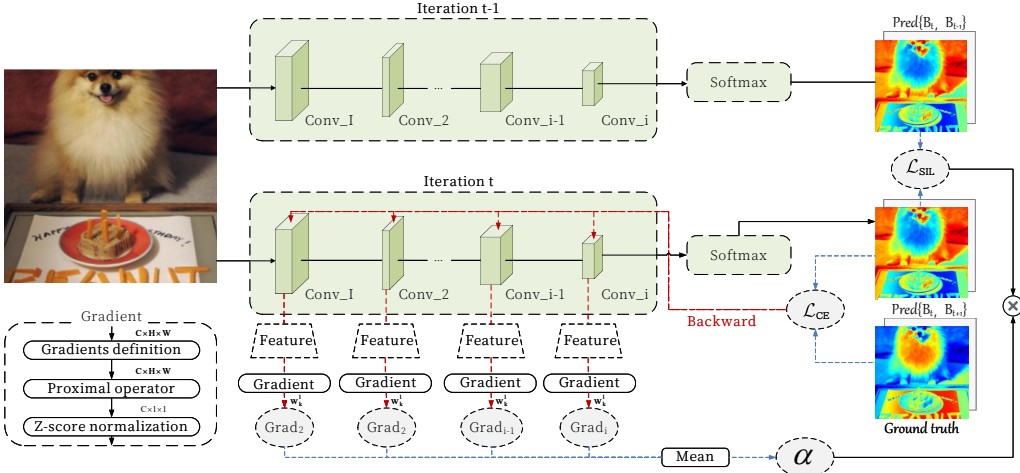

Figure 2: An overview of our SDPGO framework, which uses sequential iterative learning and gradient-based feature weighting to generate soft labels. This process calculates the KL divergence between iterations and assigns weights to features according to their gradient contributions.

- **Implementation and Evaluation:** We conduct extensive experiments on image classification, object detection, and semantic segmentation datasets based on different popular network architectures, where the proposed SDPGO framework outperforms recent state-of-the-art KD methods with a clear margin. In addition, ablation experiments further demonstrate the effectiveness of our method.

## 2 Related Works

KD compresses neural networks by transferring knowledge from a high-capacity teacher to a lightweight student, enabling efficient deployment of the compact model. KD leverages the teacher model's knowledge by utilizing 1) class predictions as soft labels [60, 7], 2) correlations between multiple samples [33, 42, 3], or 3) feature maps containing spatial information from intermediate layers [4, 2, 10]. Under the guidance of a teacher model, the student model trained in an offline learning way is typically effective. However, the entire training process is time-consuming and complex, as it requires retaining a large amount of persistent data to train a strong teacher model. In contrast, online KD [27, 52, 34, 39] eliminates the need for larger teacher models, instead training a group of student networks by learning from each other's predictions. However, *there are two key limitations: (1) online KD depends on using an ensemble teacher to improve the student model's performance, and (2) selecting the optimal student model requires considerable computational resources for training.*

Compared with the conventional KD, SKD [55, 58, 19] uses the student network itself as a teacher to guide its learning process, which utilizes the information within the student model. In recent years, a series of SKD methods have been continuously proposed. BYOT [58] attaches auxiliary classifiers to each stage, using the final classifier's output as teacher knowledge to train them. FRSKD [18] employs an auxiliary self-teacher network with bidirectional paths to transfer refined features and soft labels to the classifier. However, these methods rely on auxiliary models, which complicates training by increasing memory and time costs. Meanwhile, data augmentation was used for SKD. For example, MiSKD [51] combines SKD with image mixture to distill features and distributions between original/mixed images via a self-teacher network, enhancing self-boosting. Meanwhile, SKD uses historical information as a virtual teacher, treating predictions from previous model versions as regularization targets. These methods takes the predictions output for knowledge transfer from its last generation [9], last mini-generation [50, 36] or last epoch [20].

Recently, some researchers [25, 63] begin to employ gradients to assess the importance of feature maps. These methods assigns static or predefined weights to features impacting task loss, primarily emphasizing knowledge transfer from teacher to student. Unlike previous approaches, we design a proximal gradient optimization mechanism that dynamically refines gradient-derived weights during training. This adaptively prioritizes features most relevant to task loss, optimizing representation learning in an end-to-end manner.

# 3 Proposed Approach: SDPGO

**Notations.** We denote a set of labeled dataset as $\mathcal{D} = \{(\mathbf{x}_i, y_i)\}_{i=1}^N$, where $N$ is the dataset's size. $x_i^k$ denotes the $i$-th sample and it belongs to category $k$. We sample a batch samples to feed into target neural network $f(\cdot)$ to optimize the cross-entropy loss function. Then, we obtain the logit vector $z_i$, and then yields the prediction probabilities $\mathbf{p}_i(k) = f(x_i^k/\tau)$ by a softmax function. Hence, the temperature parameter $\tau$ is introduced to control the importance of the $i$-th soft target as follows:

$$\mathbf{p}_i(k) = \frac{\exp(z_i/\tau)}{\sum_{j=1}^K \exp(z_j/\tau)}, \tag{1}$$

where the $z_i$ is the predicted value of class $i$ from fully connected layer, and $\tau$ denotes a temperature parameter. A softmax function with $\tau$ transforms an original vector into a probability vector.

## 3.1 Sequential Iterative Learning Module

We employ the sequential iterative learning module to measure the consistency of two adjacent batches. This module uses backup information from the last mini-batch to generate soft targets. We partition each mini-batch into two sequential segments: half aligned with the prior iteration and half with the next iteration. This strategy enables real-time distillation of dynamically updated soft targets generated from preceding training steps.

The student model acts as its own teacher by reorganizing sequential sampling. Each mini-batch is divided into halves aligned with the previous and upcoming iteration respectively. The model applies KL divergence between classifier outputs from different iterations to align predictions, while utilizing historical outputs for enhanced regularization. We denote the current batch of $n$ samples in the $t^{th}$ iteration as $\mathcal{B}_t = \{(\mathbf{x}_i^t, y_i^t)\}_{i=1}^n$. Afterwards, they are fed into convolutional layers to obtain their representation vectors as prediction distributions $\mathbf{p}_i^{\tau,t}$. Our work uses historical information from the last batch to efficiently generate soft targets as more instant smoothed labels for regularization. we substitute the $\mathbf{p}_i^{\tau,t-1}$ by the soften labels $\mathbf{p}_i^{\tau,t-1}$ generated by the identical network at $t-1$-th iteration. The main difference from conventional KD is that the teacher model dynamically evolves during training, with the $t$-th iteration predictions used as the teacher's knowledge without incurring any loss. Consequently, the student is trained with the consistency regularization loss as follows:

$$\mathcal{L}_{SIL} = -\tau^2 \frac{1}{n} \sum_{i=1}^n \cdot \underbrace{\sum_{i=1}^n \mathbf{p}_i^{\tau,t-1} \log \frac{\mathbf{p}_i^{\tau,t-1}}{\mathbf{p}_i^{\tau,t}}}_{D_{KL}}. \tag{2}$$

where $\mathbf{p}_i^\tau$ is the soften prediction, parameterized by temperature $\tau$ from the student itself. Then, the total loss combines KL divergence with hard-label cross-entropy, enabling end-to-end optimization of the single network:

$$\mathcal{L}_{CE} = \frac{1}{N} \sum_{i=1}^N H(\mathrm{y}_i, \mathrm{p}_i), \tag{3}$$

where $H$ is the cross-entropy loss between the ground-truth label $y_i$ and the prediction $p_i$.

## 3.2 Proximally Weight Assignment Module

We propose a gradient-driven dynamic weighting scheme to enhance knowledge distillation, where feature importance is evaluated via gradient magnitudes. High-gradient features receive amplified attention to prioritize impactful knowledge, while low-gradient features trigger adaptive recalibration toward under-optimized parameters. To obtain the importance of the feature, we define the gradients of the $k$-th feature map in layer $l$ as:

$$w_k^l = \frac{1}{W} \sum_{i=1}^W \sum_{j=1}^H \left| \frac{\partial L_{Task}}{\partial F_{i,j,k}^l} \right| \tag{4}$$

where $F_{i,j,k}^l$ is the $k$-th feature representations of the $l$-th layer, and $i, j$ are two location vectors. Next, we calculate the gradient of the loss $L_{ask}$ with respect to the feature $F$. The raw gradient-derived weights $w_k^l$ (Eq. 4) is refined via a proximal operator to enforce sparsity and stability:

$$\hat{w}_k^l = \text{Prox}_\lambda \left( w_k^l \right) = \begin{cases} w_k^l - \lambda & \text{if } w_k^l > \lambda \\ 0 & \text{otherwise} \end{cases} \tag{5}$$

where $\lambda$ is an adaptive threshold controlling feature sparsity using only moving averages, thus the corresponding loss can be: $\lambda_t = \beta \lambda_{t-1} + (1 - \beta) \cdot \frac{|w|_1}{K}$. $\lambda_t$ is dynamic threshold at step $t$, $\beta$ is the momentum factor. $|w|_1$ is L1-norm of gradient wights in current batch. $K$ is the numbers of features in batch. This step suppresses less impactful features while retaining those with significant gradients, aligning with the proximal gradient principle of separating critical and non-critical components. Next, we compute the mean absolute value of the gradient for each weight parameter. Then, we apply Z-score normalization to these values, given by:

$$M^l = \text{Z-score} \left( \left| \sum_{k=1}^{M} \hat{w}_k^l \right| \right) \tag{6}$$

where $M^l$ is a spatial feature map ($\mathbf{R}^{H \times W}$) derived from channel-summed activations with position-wise Z-score normalization, where $H$ and $W$ represent height and width, respectively. $l$ indicates an intermediate layer. $L$ is the total number of intermediate layers considered for distillation. Dimensionality reduction occurs through spatial averaging after normalization. We assign weights to features according to their impact on overall loss. This process effectively highlights the most influential features, ensuring that the model focuses on the aspects of the data that contribute the most to its performance. The method incorporates a dynamic weight, $\alpha$, that balances the task and distillation losses. This scalar parameter is integrated into the final loss function to modulate the influence of each component. Therefore, $\alpha$ in all network layers is computed as follows:

$$\alpha = \frac{1}{L} \sum_{l=1}^{L} \left( \frac{1}{HW} \sum_{i=1}^{H} \sum_{j=1}^{W} M^l[i, j] \right) \tag{7}$$

In previous SKD methods, the student remains unchanged because the overall task loss is usually set to a fixed value during training. However, SDPGO go a step further by leveraging the gradients of feature maps at immediate layers and transforming them into a dynamic hyperparameter. Similar to learning rate scheduling, this hyperparameter controls the degree to which the model captures valuable information about each feature's contribution. This approach allows the model to dynamically adjust its focus on features as their importance changes during training. The goal of our SDPGO method is to utilize gradient information via assigning greater weights to critical features.

### 3.3 Overall for SKD

With the sequential iterative learning module and the gradient information extraction module, we calculate the corresponding loss and propose SDPGO as shown in Fig. 2. We train all models with the total loss for SKD as follows:

$$\mathcal{L}_{\text{Total}} = \mathcal{L}_{\text{Task}} + \alpha \mathcal{L}_{SIL}. \tag{8}$$

where $\mathcal{L}_{\text{Task}}$ is the domain-specific final objective loss. For image classification setting, this is defined as the cross-entropy loss between the hot ground-truth label $y$ and the predicted distribution $p$.

## 4 Experiments

In this paper, we conduct experiments on five visual recognition datasets, namely CIFAR-10/100 [24], ImageNet [6], CUB200-2011 [44], and Cars196 [23]. We employ eight representative architectures [40] for evaluation, namely VGG-16/19 [37], ResNet-32/110 (abbreviated as R-32/110) [13], WRN20-8 [56], DNet-40-12 [17], ShuffleNet-V2 (abbreviated as SN-V2) [32] and MobileNet-V2 (abbreviated as MN-V2) [16]. All results are reported in means (standard deviations) over 3 trials.

Table 1: Top-1 accuracy (%) of various SKD methods across widely used networks on CIFAR-10 (C10) and CIFAR-100 (C100). The best results are highlighted in bold, while the second-best results are underlined. We use $\Delta$ to show its performance gain.

| Dataset | Methods | Vgg-16 | R-32 | R-110 | WRN-20-8 | DNet-40-12 | SN-V2 | MN-V2 |
|---------|---------|--------|------|-------|----------|------------|-------|-------|
| C10 | Baseline | 93.97 | 93.46 | 94.79 | 94.53 | 92.91 | 92.70 | 93.31 |
| | BYOT [58] | 94.03 | 93.57 | 94.86 | 94.14 | 93.01 | 92.99 | 93.73 |
| | EFWSNet [62] | 93.85 | 93.97 | 94.92 | 94.68 | 93.39 | 93.23 | 93.88 |
| | PS-KD [20] | 94.10 | 94.04 | 94.91 | 95.01 | 93.23 | 93.45 | 94.02 |
| | FRSKD [18] | 94.38 | 94.78 | 95.23 | 95.27 | 94.21 | 94.17 | 94.76 |
| | DLB [36] | 94.62 | 94.15 | 95.15 | 95.54 | 93.43 | 95.10 | 94.46 |
| | MiSKD [51] | 93.82 | 95.59 | 95.93 | 93.93 | 94.25 | 95.29 | 94.91 |
| | FASD [48] | 94.21 | 95.45 | 95.66 | 94.52 | 94.39 | 95.34 | 94.86 |
| | SDPGO | **95.90** | **96.44** | **95.98** | **95.70** | **95.60** | **95.73** | **95.43** |
| | $\Delta$ | 1.93 | 2.98 | 1.19 | 1.17 | 2.69 | 3.03 | 2.12 |
| C100 | Baseline | 73.63 | 71.74 | 76.36 | 77.58 | 71.69 | 71.82 | 68.08 |
| | BYOT [58] | 73.79 | 72.39 | 77.75 | 77.68 | 77.04 | 72.97 | 68.72 |
| | EFWSNet [62] | 73.92 | 73.54 | 75.81 | 78.02 | 76.95 | 72.87 | 69.45 |
| | PS-KD [20] | 74.05 | 72.51 | 77.15 | 78.74 | 72.52 | 74.55 | 69.74 |
| | FRSKD [18] | 76.72 | 75.34 | 79.15 | 78.95 | 77.12 | 75.23 | 70.25 |
| | DLB [36] | 76.12 | 74.07 | 78.18 | 79.21 | 72.52 | 75.51 | 69.47 |
| | MiSKD [51] | 76.57 | 75.12 | 78.86 | 78.07 | 76.85 | 76.52 | 71.66 |
| | FASD [48] | 75.52 | 75.42 | 78.52 | 78.62 | 77.24 | 76.76 | 71.75 |
| | SDPGO | **76.85** | **75.57** | **79.31** | **79.36** | **78.04** | **77.29** | **72.25** |
| | $\Delta$ | 3.22 | 3.83 | 2.95 | 1.78 | 6.35 | 5.47 | 4.17 |

**Dataset.** **CIFAR-10/100** [24] contain a total number of 60,000 RGB natural images of $32\times32$ small RGB images drawn from 10 and 100 categories, respectively. For **ImageNet** [6], we use 1.2 million images for training and 50,000 images for validation. The size of input images after pre-processing is $224 \times 224$. **CUB200** [44] and **Cars196** [23] are used for fine-grained visual recognition (FGVR) tasks. Different from CIFAR and ImageNet, FGVR datasets typically have fewer data instances per class, making the task more challenging. More details are available in Appendix A.

**Baselines.** We compare our SDPGO with different SOTA SKD methods, be your own teacher (BYOT) [58], feature-sharing and weight-sharing based ensemble network (EFWSNet) [62], progressive self-KD (PS-KD) [20], feature refinement via SKD (FRSKD) [18], distillation from last mini-batch (DLB) [36], SKD from image mixture (MiSKD) [51], normalized knowledge distillation (NKD) [54], and feature augmentation based self-distillation method (FASD) [48].

**Implementation details.** We used the Stochastic Gradient Descent (SGD) optimizer with a momentum of 0.9, a weight decay of 5e-4, and a temperature $\tau$ of 3. On CIFAR-100, we used a batch size of 64, an initial learning rate of 0.05, and trained the model for 240 epochs. For ImageNet, training was conducted over 100 epochs, with learning rate adjustments at epochs 30, 60, and 90 by multiplying the learning rate by 0.1. For CUB200, the learning rate was set to 0.05 with a warmup period of 2 epochs, a batch size of 64, and a total of 240 training epochs. The learning rate was adjusted at epochs 70, 140, and 210 by multiplying it by 0.1, while all other hyperparameters remained constant. For Cars196, training was conducted for 200 epochs, with learning rate adjustments at epochs 60, 120, and 180 by multiplying it by 0.1. Experimental details about Pascal VOC, ADE20K and Cityscapes are in Appendix B.

### 4.1 CIFAR Classification Result

We compare the KD results of different methods under various backbone networks settings in Table 1. Across different architectures, all networks show improvements on these two datasets when using SDPGO. We first compare our method with the latest SKD techniques, BYOT [58] and EFWSNet [62], both of which are combined with auxiliary architectures. Our method consistently delivers significant improvements to the student. Specifically, ResNet-32 achieves 94.44% and 75.57% top-1 accuracy on CIFAR-10 and CIFAR-100, respectively, reflecting accuracy gains of 5.87% and 3.83% than BYOT [58] through the knowledge transferred from gradient information. Furthermore, compared to input-space data augmentation methods, our SDPGO approach provides a significant boost in top-1 accuracy for classification tasks. Specifically, with a DNet-40-12 backbone, SDPGO achieves gains of 1.35% on CIFAR-10 and 1.19% on CIFAR-100 than MixSKD. This result shows that student models distilled by SDPGO benefits from our gradient optimization as well.

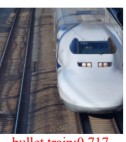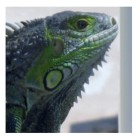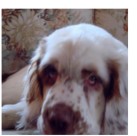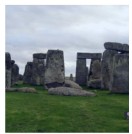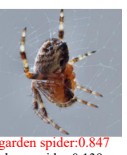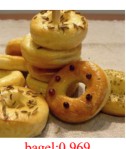

| Pomeranian:0.839 | bullet train:0.717 | common iguana:0.994 | Clumber:0.851 | megalith:0.926 | garden spider:0.847 | bagel:0.969 |
|---|---|---|---|---|---|---|
| Shetland sheepdog:0.049 | Jean:0.019 | African chameleon:0.002 | English setter:0.067 | stone wall:0.049 | barn spider:0.139 | pretzel:0.007 |
| Persian cat:0.040 | Streetcar:0.010 | green lizard:0.002 | Brittany spaniel:0.046 | castle:0.009 | spider web:0.006 | bakery:0.004 |

Figure 3: Visualization of the target and top-2 non-target class values of our customized soft labels.

Table 2: Top-1 and Top-5 accuracy (%) of our SDPGO method. ResNet-18 is used as classifier network on the ImageNet dataset.

| Methods | ResNet-18 Top-1 | ResNet-18 Top-5 | Gain($\uparrow$) |
|---|---|---|---|
| ResNet-18 | 69.75 | 89.07 | - |
| FitNet [1] | 71.61 | 90.51 | 1.86 |
| Review [4] | 70.81 | 89.98 | 1.06 |
| CAT-KD [12] | 71.22 | 90.26 | 1.47 |
| CRD [42] | 71.17 | 90.13 | 1.42 |
| SSKD [47] | 71.62 | 90.67 | 1.87 |
| DCCD [30] | 71.95 | 90.88 | 2.2 |
| BYOT [58] | 69.84 | 89.62 | 0.09 |
| EFWSNet [62] | 72.36 | 91.74 | 2.61 |
| PS-KD [20] | 71.59 | 90.85 | 1.84 |
| FRSKD [18] | 70.17 | 90.52 | 0.42 |
| DLB [36] | 70.12 | 90.27 | 0.37 |
| MiSKD [51] | 71.67 | 91.20 | 1.92 |
| FASD [48] | 71.70 | 90.91 | 1.95 |
| **SDPGO (Ours)** | **72.47** | **92.56** | **2.72** |

Table 3: Top-1 accuracy (%) of various self-knowledge distillation methods across widely used networks on CUB200 and Cars196 dataset.

| Method | CUB200 Top-1 | CUB200 Top-5 | Cars196 Top-1 | Cars196 Top-5 |
|---|---|---|---|---|
| ResNet-18 | 69.66 | 91.66 | 71.82 | 91.04 |
| + BYOT [58] | 73.38 | 91.18 | 79.35 | 94.70 |
| + DLB [36] | 76.10 | 93.37 | 78.28 | 93.13 |
| + MiSKD [51] | 71.11 | 91.37 | 82.94 | 95.83 |
| + SDPGO | **78.06** | **94.77** | **84.17** | **96.44** |
| ResNet-50 | 74.36 | 92.52 | 76.44 | 92.26 |
| + BYOT [58] | 77.76 | 94.22 | 80.17 | 94.78 |
| + DLB [36] | 80.69 | 95.62 | 82.94 | 95.83 |
| + MiSKD [51] | 75.96 | 93.03 | 77.90 | 93.55 |
| + SDPGO | **81.69** | **95.88** | **89.20** | **98.16** |
| MobNet-V2 | 73.09 | 91.82 | 72.15 | 93.01 |
| + BYOT [58] | 74.25 | 91.92 | 82.51 | 95.03 |
| + DLB [36] | 78.08 | 94.32 | 81.72 | 92.93 |
| + MiSKD [51] | 73.65 | 91.23 | 82.69 | 94.41 |
| + SDPGO | **78.67** | **94.75** | **85.28** | **96.69** |

## 4.2 ImageNet Classification Result

As shown in Table 2, SDPGO outperforms advanced KD methods in terms of top-1 and top-5 accuracies, achieving the highest scores of 72.47% and 92.56%, respectively. We select current feature-based KD methods with the utilization of intermediate representation or feature embedding. For feature-based methods, the second-best method DCCD [30] exhibits inferior performance to our SDPGO, highlighting its limitations in distilling knowledge due to relying on expert knowledge from the teacher model. For self-KD methods, SDPGO surpasses the best-competing FASD [48] by 0.77% top-1 accuracy gains, which enables successful SKD. SDPGO surpasses other SOTA methods with clear gains, demonstrating its superiority in large-scale datasets. These findings confirm the effectiveness of SDPGO in distillation optimization, highlighting its versatility and strength.

## 4.3 Fine-grained Classification Result

For fine-grained image classification, Table 3 presents the top-1 and top-5 accuracy of SDPGO compared to other SKD methods across different backbones. We observe the top-1 acc of 82.94% and 82.69% on Cars196 when Mixup is combined with SKD, which is 4.66% and 0.97% improvement of DLB. When comparing with MixSKD, SDPGO demonstrates superior performance on fine-grained classification tasks, achieving the highest accuracy across all datasets. Specifically, SDPGO surpasses the second-best results by 1.60%, 3.63%, and 1.59% on three networks, respectively. This significant improvement highlights the effectiveness of our approach in combining sequential iterative learning and gradient information extraction.

## 4.4 Visualization

Figure 3 illustrates how our self-distillation method, SDPGO, generates customized soft labels for each image during training, including the target class and the top two non-target classes. By using the gradient-based dynamic weight assignment method from Eq. 6, higher weights are assigned to classes similar to the target, enhancing the overall learning process. These results demonstrate that SDPGO's soft labels prioritize classes with high semantic affinity, thereby facilitating improved discrimination and robustness by emphasizing inter-class relationships during training.

Table 4: The downstream tasks results with SDPGO on the COCO-2017 dataset.

| Methods | ResNet-18 | | ResNet-50 | |
|---------|-----------|------|-----------|------|
| | bbox | segm | bbox | segm |
| baseline | 33.4 | 30.2 | 36.9 | 33.4 |
| MixSKD | 33.9 | 31.05 | 37.0 | 33.8 |
| FASD | 34.1 | 30.9 | 37.3 | 34.4 |
| SDPGO | **35.08** | **32.10** | **38.08** | **36.69** |

Table 5: Results of training more models, including ViT-liked models with SDPGO on ImageNet.

| Model | Baseline | SDPGO (Ours) |
|-------|----------|--------------|
| ResNet-50 | 73.56 | **74.03** (+0.47) |
| DeiT-Tiny | 74.42 | **75.01** (+0.59) |
| DeiT-small | 80.55 | **80.89** (+0.34) |
| Swin-Tiny | 81.18 | **81.95** (+0.77) |
| Swin-small | 84.36 | **86.24** (+1.88) |

Table 6: Overall performance comparison on semantic segmentation tasks.

| Model | Method | ADE20K | Cityscapes |
|-------|--------|--------|------------|
| | Baseline | 39.72 | 74.85 |
| | MixSKD | 42.37 | 74.96 |
| ResNet-50 | FASD | 40.78 | 72.89 |
| | SDPGO | **42.75** | **75.01** |

Table 7: Performance comparison on Pascal VOC segmentation task.

| Model | Method | mIOU | Model | Method | mIOU |
|-------|--------|------|-------|--------|------|
| | Baseline | 79.07 | | Baseline | 81.95 |
| EfficientDet-d0 | MixSKD | 79.52 | EfficientDet-d1 | MixSKD | 82.51 |
| | FASD | 80.54 | | FASD | 83.43 |
| | SDPGO | **80.67** | | SDPGO | **83.97** |

## 4.5 Extension to ViT-liked models

Recent models like ViT [8], which use embedded image patches, have adopted a different architecture, but current SKD methods do not address their unique characteristics. SDPGO leverages gradient information to prioritize features, eliminating the need for additional linear layers connected to ViT's middle layer, and allowing the model to focus on the most relevant aspects during training. As can be seen in Table 5, SDPGO outperforms these baselines by margins ranging from $0.34\% \sim 1.88\%$. It should be noted that our method even improves student model performance with gains of up to 1.88% for the latest state-of-the-art model Swin-Transformer (Swin-small) [31]. These results show that SDPGO achieves significant improvements with general settings.

## 4.6 Knowledge transfer to downstream tasks

To assess its generalization capability, we transfer our model to object detection task on the COCO-2017 dataset [29]. As shown in Table 4, ResNet trained with SDPGO brings the feature extractors with average 1.43% and 2.60% points for downstream detection task compared to the baseline method. The results demonstrate the efficacy of SDPGO for learning better representations for downstream semantic recognition tasks. In addition, we employ the Pascal VOC, ADE20K, and Cityscapes benchmarks for semantic segmentation. As shown in Table 6, SDPGO achieves the best segmentation performance and outperforms the best-second results with 2.65% and 3.04% mIoU margins on Pascal ADE20K and Cityscapes segmentation, respectively. For Pascal VOC 2012 semantic segmentation, we use EfficientDet with stacked BiFPN structure [41] as a baseline. Table 7 demonstrates that SDPGO substantially outperforms baseline segmentation models by dynamically intensifying task-critical features via gradient-weighted distillation. Empirical results reveal that the proposed method generalizes remarkably well to other domains like object detection and segmentation without requiring architectural changes.

## 5 Analysis

In this section, we provide an in-depth analysis of SDPGO. We begin with a theoretical justification behind the efficacy of SDPGO. Then we perform a qualitative analysis to visualize the representations of the models. Finally, we explore the impact of different design choices through ablation studies, focusing on the effects of utilizing the gradient optimization.

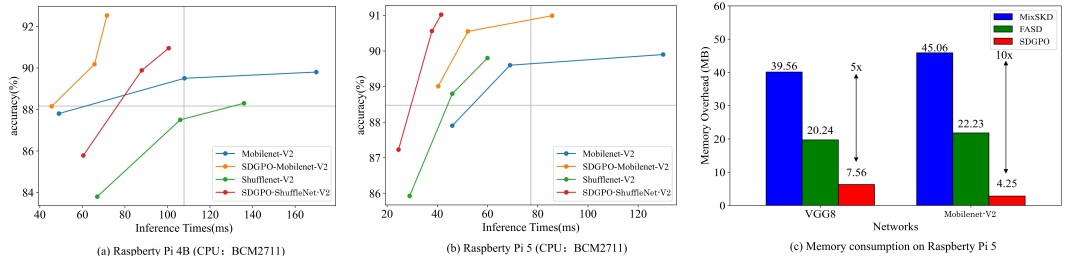

(a) Raspberty Pi 4B (CPU: BCM2711)  (b) Raspberry Pi 5 (CPU: BCM2711)  (c) Memory consumption on Raspberty Pi 5

Figure 4: Inference time and memory consumption on ImageNet. (a) and (b) represent the model tested on Raspberry Pi 4B and Raspberry Pi 5, respectively. In (c), "Memory" indicates whether additional information needs to be stored during knowledge distillation.

Table 8: Top-1 Acc (%) and training time for each batch of data of competitive KD methods.

| Method | Baseline | ReviewKD | CATKD | BYoT | PS-KD | DLB | FASD | Ours |
|---|---|---|---|---|---|---|---|---|
| Training Time | **12** | 25 | 17 | 19 | 15 | 17 | 40 | **12** |
| Top-1 Acc (%) | 69.66 | 76.58 | 72.6 | 73.38 | 75.43 | 76.10 | 75.43 | **78.06** |

## 5.1 Theoretical Justification

First, we provide a theoretical foundation for the proposed Self-Distillation training via Proximal Gradient Optimization (SDPGO). The gradient of the total loss with respect to model parameters $\theta$ is:

$$\frac{\partial \mathcal{L}_{\text{Total}}}{\partial \theta} = \frac{\partial \mathcal{L}_{\text{Task}}}{\partial \theta} + \alpha \frac{\partial \mathcal{L}_{SIL}}{\partial \theta}. \tag{9}$$

By dynamically adjusting $\alpha$, SDPGO modulates the influence of self-distillation on parameter updates, ensuring stable training.

**Stability.** The sequential iterative learning module incorporates proximal gradient optimization to refine feature weights and stabilize training. Raw gradient-derived weights $\hat{w}_k^l$ are processed via a proximal operator: $\hat{w}_k^l = \text{Prox}_\lambda \left( w_k^l \right)$. $\lambda$ enforces sparsity by suppressing non-critical features. This mitigates noise amplification during self-distillation. We add the proximal term $\|\theta_t - \theta_{t-1}\|_2^2$ to the loss to penalize abrupt parameter changes:

$$\mathcal{L}_{\text{Total}} = \mathcal{L}_{\text{Task}} + \alpha \mathcal{L}_{SIL} + \beta \|\theta_t - \theta_{t-1}\|_2^2. \tag{10}$$

This ensures smooth convergence by aligning parameter updates with historical states.

**Convergence Guarantee.** To establish the convergence of SDPGO, we leverage the properties of proximal gradient descent under standard convexity and smoothness assumptions. Let the total loss $\mathcal{L}_{\text{Total}} = \mathcal{L}_{\text{Task}} + \alpha \mathcal{L}_{SIL}$ be $L$-smooth (i.e., $\|\nabla \mathcal{L}_{\text{Total}}(\theta_1) - \nabla \mathcal{L}_{\text{Total}}(\theta_2)\| \le L \|\theta_1 - \theta_2\|$ for all $\theta_1, \theta_2$) and convex. The optimization proceeds via the following iterative steps: 1. Gradient Step: $\theta_{t+1/2} = \theta_t - \eta \nabla_\theta \mathcal{L}_{\text{Total}}(\theta_t)$. 2. Proximal Step: $\theta_{t+1} = \text{Prox}_{\eta\lambda}(\theta_{t+1/2}) = \arg\min_\theta \left( \frac{1}{2\eta} \|\theta - \theta_{t+1/2}\|^2 + \lambda \|\theta - \theta_{t-1}\|^2 \right)$. If the learning rate satisfies $\eta \le \frac{1}{L}$, the sequence $\{\theta_t\}$ generated by SDPGO converges to a global minimum.

## 5.2 Efficiency and Memory Consumption Analysis.

We measure the actual speedup of lightweight neural networks on two mobile devices, Raspberry Pi 4B and Raspberry Pi 5 in Figure 4. In Figure 4 (a) and (b), SDPGO achieves an average inference acceleration optimization of 23.46% on Shuffle-V2 and Mobile-V2 backbone. For Shuffle-V2 ($\times 0.5$) on the Raspberry Pi 4B, the proposed method reduces hardware inference time by 17.08% while improving the baseline by 2.38%. Meanwhile, the average inference latency of SDPGO on the Mobile-V2 model is 60.95 ms, and the improvements is up to 34.69% on ImageNet. In Figure 4 (c), SDPGO delivers significant memory compression, achieving approximately $3\times$ reduction compared to the state-of-the-art FASD method and $5\times$ reduction compared to MixSKD on VGG-8. Moreover, we compare the training time of various KD methods in Table 8. In particular, our method shows competitive performance by assessing the training time of each batch of data on CUB200.

Table 9: Performance comparison when a fraction of training data is noisy.

| $\eta$ | BYOT | PS-KD | DLB | FASD | Ours |
|---|---|---|---|---|---|
| 0 | 72.39 | 72.51 | 74.07 | 75.42 | **75.57** |
| 10 | 65.41 | 62.75 | 67.56 | 64.25 | **71.56** |
| 20 | 58.05 | 57.56 | 65.17 | 60.51 | **68.53** |
| 30 | 53.25 | 51.71 | 54.45 | 57.27 | **61.58** |
| 40 | 42.22 | 41.26 | 51.25 | 55.39 | **59.81** |

Table 10: Performance comparison between SOTA SKD and SDPGO when a fraction of data present for training purpose.

| $F$ | BYOT | PS-KD | DLB | FASD | Ours |
|---|---|---|---|---|---|
| 25 | 49.57 | 48.75 | 51.28 | 60.34 | **65.29** |
| 50 | 58.25 | 56.23 | 59.56 | 68.62 | **70.07** |
| 75 | 63.43 | 60.58 | 68.12 | 70.47 | **71.58** |
| 100 | 72.39 | 72.51 | 74.07 | 75.42 | **75.57** |

## 5.3 Robustness Analysis under Noisy and Scarce Data

We experiment with our method for few-shot and noisy-label learning on CIFAR-100. We evaluate SDPGO against standard KD baselines under symmetric label noise (uniform corruption probability $\eta$). Experiments use ResNet-32 with $\eta \in 0\%$, 10%, 20%, 30%, and 40%. As summarized in Table 9, the proposed SDPGO proves to be the most resilient to symmetric label noise. At a noise level of $\eta$=40%, it attains a top accuracy of 59.81%, outperforming the strongest competitor, FASD (55.39%), by a clear margin. For few-shot learning, we train ResNet-32 on random subsets of CIFAR-100, using 25%, 50%, 75%, and 100% of the training data per class. The results in Table 10 validate the sample efficiency of SDPGO, which achieves the highest top-1 accuracy across all data fractions (e.g., 65.29% at 25%), markedly outperforming conventional SKD methods.

## 5.4 Integration Analysis

SDPGO's modular architecture enables seamless compatibility with diverse self-distillation (SKD) paradigms. We evaluate our pre-process on existing KD methods. These findings indicate that using SDPGO as a plugged-in regularization can enhance the generation of other approaches. As shown in Table 11, student models distilled by SDPGO benefits from our pre-process as well.

Table 11: The results of SDPGO combined with other advanced SKD techniques on CIFAR-100.

| Model | VGG-13 | ResNet-32 | ResNet-110 | ShuffleNet-V2 | MobileNet-V2 |
|---|---|---|---|---|---|
| KD | 72.98 | 73.08 | 74.36 | 71.82 | 66.95 |
| KD+Ours | **73.53** | **74.32** | **75.11** | **72.72** | **67.24** |
| DLB | 75.45 | 74.07 | 78.18 | 75.51 | 69.47 |
| DLB+Ours | **75.67** | **75.19** | **79.28** | **77.56** | **69.82** |

More ablation studies can be found in Appendix C.

## 6 Conclusion

In this work, we introduce a framework called Self-Distillation training via Gradient Optimization (SDPGO). This framework leverages gradient information to prioritize features that significantly influence classification performance, allowing the model to concentrate on the most relevant aspects during training. Furthermore, it reduces computational costs while improving the model's ability to represent features. The dynamic weight adjustment scheme refines the gradient information into a variable weighting factor to evaluate the distillation knowledge. Our SDPGO method effectively enhances the student model's generalization capability by emphasizing key features linked to classification loss. Extensive experiments on five image classification benchmark datasets at different scales show that our framework outperforms recent state-of-the-art knowledge distillation techniques.

## Acknowledgment

This work is partially supported by NSFC (Natural Science Foundation of China): 61602345,62002263; National Key Research and Development Plan: 2019YFB2101900; Key Project of Tianjin Natural Science Foundation: 25JCZDJC00250. TianKai Higher Education Innovation Park Enterprise R&D Special Project: 23YFZXYC00046. The Tianjin Science and Technology Program under Grant: 24YDTPJC00630, Tianjin Municipal Education Commission Research Program Project under No. 2022KJ012; Foundation of Key Laboratory of Big Data & Artificial Intelligence in Transportation (Beijing Jiaotong University), Ministry of Education (No.BATLAB202401).

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

# A    Supplement to Method

## A.1    Datasets

**The MS-COCO dataset [29]** has been established as a large-scale benchmark and a de facto standard for advancing the state of the art in object detection and instance segmentation. Its curated collection consists of 118,000 images for model training and 5,000 images for validation. Each image is meticulously annotated with precise bounding boxes and per-instance segmentation masks, encompassing a diverse set of 80 commonly encountered object categories.

**The ADE20K dataset [64]** serves as a benchmark for semantic segmentation and scene parsing, comprising 20,210 training, 2,000 validation, and 3,352 testing images. It is characterized by its fine-grained annotation of 150 object and "stuff" categories, presenting a challenging testbed for holistic scene understanding models due to its complex scenes and dense, pixel-level labels.

**The Cityscapes dataset [5]** is a benchmark for urban scene understanding, particularly in autonomous driving. It comprises 5,000 high-resolution images with fine, pixel-level annotations and 20,000 images with coarse annotations. The finely annotated set is divided into 2,975 training, 500 validation, and 1,525 testing images. While annotations span 30 classes, 19 are used for standard evaluation. This two-tiered annotation structure supports both precise model training and large-scale pre-training, establishing Cityscapes as a vital resource for semantic segmentation in complex urban environments.

## A.2    Evaluation Metrics

For multi-class classification, we use top-1 and top-5 acc as standard performance measures. Meanwhile, the complexity is evaluated by the sum of floating point operations (FLOPs) in one forward on a fixed input size. To evaluate the segmentation accuracy, we adopt the mean Intersection-over-Union (mIoU) as the primary performance metric. This provides an aggregate measure of segmentation accuracy across all classes.

# B    Experimental Setups

We evaluate SDPGO on the PASCAL VOC dataset using Faster-RCNN [35] object detection framework. The backbone architecture utilized in this framework is EfficientDet-d0/d1 [41]. For these tasks, we employ the ADE20K [64] and Cityscapes [5] datasets and use ResNet-50 as the segmentation model. we adhere to the training protocol and hyperparameter settings established in [53] to ensure a fair and reproducible comparison. All experiments are conducted on $8 \times$ NVIDIA Tesla-A100 GPUs.

# C    More Experiments

## C.1    Efficiency and Memory Consumption

In this section, we conduct experiments on the efficiency and memory consumption on CUB200, as shown in Table 12. The flops and memory are selected as an evaluation metric to validate the proposed method. $T$ represents introducing additional overhead, while $F$ represents not introduce additional overhead. Resnet-18 is selected as the backbone network. Compared with other state-of-the-art approaches, our proposed SDPGO method achieves the highest top-1 accuracy of 78.06% while maintaining minimal computational overhead during both inference and training stages. Specifically, during inference, all methods utilize the same number of parameters (11.3M) and Flops (1.82G), but SDPGO outperforms others in accuracy by a significant margin. For instance, methods like BYOT, EFWSNet, PS-KD, FRSKD, and MiSKD yield lower accuracies ranging from 71.11% to 73.38%, despite similar inference costs.

In the training phase, SDPGO requires only 1.82G Flops and does not incur additional memory consumption (denoted by "F"), demonstrating high efficiency. In contrast, several methods introduce substantial computational burdens. For example, MiSKD increases Flops to 7.30G and uses extra memory ("T"), while BYOT and PS-KD also require higher Flops (5.75G and 3.65G, respectively) and memory overhead. Although FASD and DLB maintain low Flops (1.84G and 1.82G) and avoid memory issues, their accuracies (75.43% and 76.10%) are lower than SDPGO. This highlights

that SDPGO effectively balances accuracy and efficiency, avoiding the need for extensive data augmentations or auxiliary components that escalate training costs, as seen in methods like MiSKD or BYOT. Thus, SDPGO provides a superior solution for resource-constrained scenarios without compromising performance.

Table 12: Comparison Of Efficiency and Memory Consumption on CUB-200.

| Method | Inference Stage | | | Training Stage | | |
|--------|--------|-------|-----------|--------|-------|--------|
| | Params | Flops | Top-1 acc | Params | Flops | Memory |
| BYOT [58] | 11.3M | 1.82G | 73.38 | 11.3M | 5.75G | T |
| EEWSNet [62] | 11.3M | 1.82G | 72.97 | 11.3M | 3.64G | F |
| PS-KD [20] | 11.3M | 1.82G | 72.65 | 11.3M | 3.65G | T |
| FRSKD [18] | 11.3M | 1.82G | 73.14 | 11.3M | 3.05G | F |
| DLB [36] | 11.3M | 1.82G | 76.10 | 11.3M | 1.82G | F |
| MiSKD [51] | 11.3M | 1.82G | 71.11 | 11.3M | 7.30G | T |
| FASD [48] | 11.3M | 1.82G | 75.43 | 11.3M | 1.84G | F |
| SDPGO | 11.3M | 1.82G | 78.06 | 11.3M | 1.82G | F |

## C.2 More Ablation Studies

**Ablation study on Hyper-parameter** $\lambda$**.** In this section, we compare hard and soft thresholds on CUB200. Our method still shows competitive performance when using adaptive threshold strategy. As shown in the Table 13, the conventional knowledge distillation (KD) baseline ("N/A") achieves 71.74% and 71.82% on ResNet-32 and ShuffleNet-V2, respectively. In contrast, both our fixed threshold and adaptive threshold strategies bring substantial improvements. Specifically, the fixed threshold version attains 75.57% with ResNet-32 and 77.29% with ShuffleNet-V2, while the adaptive threshold further boosts performance to 75.78% and 77.63%, respectively. Moreover, the adaptive threshold strategy, despite its simplicity, consistently outperforms the fixed variant across both architectures. This validates the effectiveness and architectural flexibility of our proposed thresholding approach in knowledge distillation.

**Ablation study on loss function changes of SDPGO.** The impact of different loss functions on our method is quantitatively evaluated on CIFAR-100, with results detailed in Table 14. The core mechanism of SDPGO, dynamic feature weighting via proximal gradients, depends solely on gradient amplitudes and not on the semantics of the loss function. We tested SDPGO with three distinct losses on CIFAR-100. Whether $\mathcal{L}_{\text{Task}}$ is cross entropy or focal loss, $|\nabla_\theta \mathcal{L}|$ indicates features critically impacting task performance. As shown in Table 14, the top-1 acc of SDPGO obtain the performance gain compared with the baseline when the the loss function changes. Specifically, When the standard cross-entropy loss is employed, SDPGO achieves a top-1 accuracy of 77.29%, significantly outperforming the corresponding baseline of 71.82%. This demonstrates a substantial performance gain of 5.47% attributable to our proposed framework under the conventional loss setting. In contrast, when utilizing Focal Loss, the performance of both the baseline and SDPGO decreases. The baseline drops to 69.23%, and SDPGO attains 72.36%, resulting in a narrower margin of improvement (3.13%).

Table 13: Performance of the SDPGO method using hard/soft thresholds. 'N/A' denotes the result of the conventional KD.

| Method | ResNet-32 | ShuffleNet-V2 |
|--------|-----------|---------------|
| N/A | 71.74 | 71.82 |
| Fixed | 75.57 | 77.29 |
| Adaptive | **75.78** | **77.63** |

Table 14: Different loss function with accuracy on CIFAR-100.

| Loss type | method | Top-1 acc |
|-----------|--------|-----------|
| Cross entropy | Baseline | 71.82 |
| | SDPGO | **77.29** |
| Focal loss | Baseline | 69.23 |
| | SDPGO | **72.36** |

