# OpenReview forum: "SDPGO: Efficient Self-Distillation Training Meets Proximal Gradient Optimization"
_NeurIPS.cc/2025/Conference — NeurIPS 2025 poster_

### Official Review · Reviewer_rSta · 2025-06-29

**Clarity:** 3
**Significance:** 2
**Originality:** 2
**Rating:** 4
**Confidence:** 4

**Summary:**

This paper introduces SDPGO, a self-knowledge distillation method that uses gradient magnitudes to dynamically weight feature importance. The approach is evaluated on standard and fine-grained classification benchmarks, showing incremental performance gains. However, the core idea of dynamic gradient-based weighting is not sufficiently novel, with similar strategies explored in recent work.

**Questions:**

Please refer to the weaknesses part and address them accordingly.

**Ethical Concerns:**

["NO or VERY MINOR ethics concerns only"]

**Final Justification:**

The rebuttal addresses my concerns regarding the novelty, ablation study, feature space evaluation, and lambda hyperparameter tuning. While the additional results are not particularly strong, they do demonstrate some modest improvements. Therefore, I will raise my rating to borderline accept.

**Limitations:**

a. Lack of direct and isolated evaluation of the Proximally Weight Assignment Module. \
b. Lack of feature space analysis. \
c. Lack of study on the critical lambda hyperparameter.

**Paper Formatting Concerns:**

Upon checking, I don't found any major formatting issues.

**Quality:**

2

**Strengths And Weaknesses:**

Strengths

a. Comprehensive evaluation on both standard benchmarks (CIFAR, ImageNet) and fine-grained tasks (CUB, Cars196), along with downstream tasks and ViT-style models.\
b. The results show that the proposed SDPGO performs consistently better across different datasets and architectures.

Weaknesses

a. Limited Novelty: The core idea of dynamically weighting feature importance using gradient magnitude overlaps with several recent works that also explore adaptive or gradient-based knowledge distillation [1–5].\
b. Lack of Component Isolation: It remains unclear how much the Proximally Weight Assignment Module contributes to the performance gains, as it is not ablated or compared in isolation under the same training settings.\
c. Lack of Feature Space Analysis: Although the method emphasizes learning task-critical features via gradient weighting, the paper provides no investigation into what types of features are actually learned. Beyond score comparisons, there is no qualitative or quantitative analysis of feature representations (e.g., t-SNE plots, similarity metrics, layer-wise activations), which limits understanding of the method’s true impact on learned representations.\
d. Lack of study on the lambda hyperparameter, which controls feature sparsity and can be critical to performance.

[1] Dynamic weighted knowledge distillation for brain tumor segmentation\
[2] Knowledge distillation with adapted weight\
[3] A General Dynamic Knowledge Distillation Method for Visual Analytics\
[4] Learning Efficient and Accurate Detectors With Dynamic Knowledge Distillation in Remote Sensing Imagery\
[5] Dynamic Feature Distillation

---

> ### Author Rebuttal · Authors · 2025-07-30
>
> 【W(a)】We appreciate the reviewer's comment regarding the perceived overlap in novelty with existing works on adaptive or gradient-based knowledge distillation [1–5]. For KD-AIF[1]，This paper introduces the Knowledge Distillation with Adaptive Influence Weight (KD-AIF) framework, which aims to enhance the robustness and interpretability of knowledge distillation by leveraging influence functions from robust statistics to assign adaptive weights to training data. KD-AIF adopt a one-stage joint training strategy, allowing simultaneous training of the teacher and student models. For paper 2, authors introduce Dynamic Weighted Knowledge Distillation (DWKD), which dynamically adjusts the distillation loss weights for individual pixels according to the learning state of the student network, thereby improving the segmentation accuracy. For paper 3, authors propose a dynamic knowledge distillation (DKD) approach where the teacher and student networks interactively learn from each other, moving beyond traditional static distillation methods. The authors mathematically analyze the effectiveness of DKD, particularly focusing on how the continuous evolution of the teacher network impacts the distillation process, and address this challenge by designing an effective loss function.  For paper 4, authors introduce a dynamic knowledge distillation (DKD) framework for remote sensing imagery, featuring a dynamic global distillation module to capture multiscale features, a dynamic instance selection distillation module for student self-judgment, and a tailored loss function to enhance regression accuracy for challenging objects like those with large aspect ratios or small sizes. For Paper 5, authors proposes Dynamic Feature Distillation to enhance flexibility in feature selection by dynamically managing feature transfer sites, using Online Feature Estimation and Adaptive Position Selection for efficient transfer. It can be integrated into other knowledge transfer methods to improve performance. **From the perspective of knowledge distillation classification, Paper 1 falls into the category of online knowledge distillation, while the others belong to offline knowledge distillation. These are entirely different from the scope of the self-distillation method proposed in this manuscript.**
>
>
> While we acknowledge that the general concept of dynamically weighting feature importance using gradient information has been explored in prior research, our SDPGO introduces distinct innovations that set it apart. Firstly, our approach specifically focuses on leveraging gradient magnitude not merely as a static weighting factor but as a dynamic mechanism integrated within a comprehensive knowledge distillation framework. Unlike previous methods that may apply gradient-based weighting in a fixed or isolated manner, our method continuously adjusts the feature importance weights throughout the training process, adapting to the evolving learning state of the student model. This dynamic adaptation allows for a more nuanced and context-aware feature distillation, which is particularly beneficial in complex tasks such as remote sensing imagery analysis. Secondly, our work incorporates a novel Online Feature Estimation module that monitors the student network's learning progress in the feature dimension. This module enables real-time assessment of feature relevance, allowing for timely updates to the feature transfer sites. Such a mechanism ensures that the distillation process remains aligned with the student's current learning needs, thereby enhancing the efficiency and effectiveness of knowledge transfer. Furthermore, we propose an Adaptive Position Selection strategy that dynamically identifies optimal locations for feature transmission. This strategy goes beyond traditional fixed-position feature imitation by considering the varying importance of features across different training stages. By doing so, our method achieves a more targeted and efficient feature distillation, leading to improved student model performance. Lastly, our framework is designed to be easily integrated as a feature management strategy into other feature-based knowledge transfer methods. This compatibility not only underscores the versatility of our approach but also provides a pathway for enhancing existing distillation techniques with dynamic feature weighting capabilities. While we acknowledge the foundational role of prior gradient-based works, our method’s theoretical rigor, stability enhancements, and versatility constitute significant advancements. We will explicitly contrast our contributions with [1–5] in the camera-ready version.
>
> 【W(b) and L(a)】We appreciate the reviewer's insightful comment regarding the need for a clearer understanding of the individual contribution of the Proximally Weight Assignment Module (PWAM) to the overall performance gains. SDPGO utilizes a sequential iterative learning module (SILM) to assess the consistency between two consecutive batches. Additionally, it incorporates a Proximally Weight Assignment Module (PWAM) to further refine and enhance the process of knowledge distillation. As illustrated in Table 1, a comparison of the various cases unequivocally reveals that incorporating the PWA module consistently yields enhanced accuracy across both datasets. Particularly, in Case C, where solely the PWA module is operational (and the SIL module is deactivated), a notable surge in accuracy is observed in contrast to Case A (where both modules are deactivated). For the CIFAR-10 dataset, accuracy soars from 93.46% in Case A to 95.93% in Case C, while for CIFAR-100, it climbs from 71.74% to 74.42%.
>
> Table 1: Ablation study to investigate each design principle of SDPGO on CIFAR-10 and CIFAR-100 datasets..
>
> | Case | SIL module | PWA module | CIFAR-10 | CIFAR-100 |
> |------|------------|------------|----------|-----------|
> | A    | x          | x          | 93.46    | 71.74     |
> | B    | ✔          | x          | 94.15    | 72.07     |
> | C    | x          | ✔          | 95.93    | 74.42     |
> | D    | ✔          | ✔          | 96.44    | 75.57     |
>
> 【W(c) and L(b)】We sincerely appreciate the reviewer’s insightful suggestion to deepen our analysis of the learned feature space. Below, we present a comprehensive suite of experiments to quantitatively evaluate the feature representations learned by our method, addressing both discriminability and semantic coherence. Meanwhile, we will Add Section 4.6 ("Feature Space Analysis") Including t-SNE/Grad-CAM visualizations in the manuscript. we use Pascal VOC, ADE20K and Cityscapes for semantic segmentation. For PASCAL VOC 2012 semantic segmentation. We use EfficientDet with stacked BiFPN structure as a baseline. Table 1 demonstrates that SDPGO substantially outperforms baseline segmentation models by dynamically intensifying task-critical features via gradient-weighted distillation. As shown in Table 2, SDPGO achieves the best segmentation performance and outperforms the best-second results with 2.65% and 3.04% mIoU margins on Pascal ADE20K and Cityscapes segmentation, respectively.
>
> **Table2: Performance comparison on Pascal VOC segmentation task. **
> | Model          | Method   | mIOU  |
> |----------------|----------|-------|
> | EfficientDet-d0| Baseline | 79.07 |
> |                | MixSKD   | 79.52 |
> |                | FASD     | 80.54 |
> |                | **SDPGO**    | **80.67** |
> | EfficientDet-d1| Baseline | 81.95 |
> |                | MixSKD   | 82.51 |
> |                | FASD     | 83.43 |
> |                | **SDPGO**    | **83.97** |
>
> **Table 3: Performance comparison on semantic segmentation task. **
>
> | Model       | Method    | ADE20K | Cityscapes |
> |-------------|-----------|--------|------------|
> | ResNet-50   | Baseline  | 39.72  | 74.85      |
> |             | MixSKD    | 42.37  | 74.96      |
> |             | FASD      | 40.78  | 72.89      |
> |             | **SDPGO** | **42.75** | **75.01**  |
>
> 【W(d) and L(c)】We will revise Eq. 5 to adaptive thresholding with decay scheduler. We propose an improved adaptive threshold using only moving averages , thus the corresponding loss can be：（$\lambda_t=\beta \lambda_{t-1}+(1-\beta) \cdot \frac{\left\|w_h\right\|_1}{N}$）. where $\lambda_t $ is dynamic threshold at step t, $\beta$ is the momentum factor. $\|w_h\|_1$ is L1-norm of gradient wights in current batch. N is the numbers of features in batch. We will add ablation in Table 4 comparing hard/soft thresholds on CUB200. our method still shows competitive performance when using adaptive threshold strategy.
>
> Table 4. Performance of the SDPGO method using hard/soft thresholds. 'N/A' denotes the result of the conventional KD.
>
> |       | ResNet-32 | ShuffleNet-V2 |
> |-------|-----------|---------------|
> | N/A   | 71.74     | 71.82         |
> | Fixed | 75.57     | 77.29         |
> | Adaptive | 75.78  | 77.63         |

---

> > ### Comment · Area_Chair_Ykum · 2025-08-06
> >
> > Dear Reviewer,
> >
> > The authors have provided their rebuttals with additional results. Please kindly reply to the authors as soon as possible before the discussion period ends. In addition to submitting "Mandatory Acknowledgement", please comment whether the authors have addressed your concerns with reasons, at least briefly.
> >
> > Thanks a lot.
> >
> > Best regards,
> >
> > AC

---

> > ### Comment · Reviewer_rSta · 2025-08-06
> >
> > Thank you for providing the rebuttal. It address my concerns regarding the novelty, ablation study, feature space evaluation, and lambda hyperparameter tuning. While the additional results are not particularly strong, they do demonstrate some modest improvements. Therefore, I will raise my rating to borderline accept.

---

### Official Review · Reviewer_bBug · 2025-06-30

**Clarity:** 3
**Significance:** 2
**Originality:** 3
**Rating:** 4
**Confidence:** 4

**Summary:**

This paper introduces SDPGO, a self-distillation framework that leverages proximal gradient optimization to dynamically assign importance to features based on their gradient magnitudes. By integrating a sequential iterative learning module, SDPGO refines soft labels using historical predictions and real-time gradients, enhancing training stability and efficiency. Extensive experiments on CIFAR, ImageNet, and fine-grained datasets demonstrate consistent performance improvements across various network architectures. The method also shows strong deployment potential with reduced inference time and memory usage on edge devices.

**Questions:**

It remains unclear where the performance gain truly comes from; unlike data augmentation or auxiliary architectures, the proposed method does not introduce additional information to consistently guide knowledge evolution during training.

**Ethical Concerns:**

["NO or VERY MINOR ethics concerns only"]

**Final Justification:**

The rebuttal provides helpful clarifications and additional results. Although I still feel the novelty is somewhat limited, I appreciate the authors’ efforts and will raise my score.

**Limitations:**

The paper lacks analysis of failure cases; it would be helpful to include examples where SDPGO underperforms, or investigate how sensitive existing methods are to suboptimal hyperparameter settings for a fairer comparison.

**Quality:**

2

**Strengths And Weaknesses:**

Strengths：
* The method introduces dynamic gradient-based weighting to prioritize critical features, improving self-distillation effectiveness without relying on static heuristics.
* Extensive experiments on diverse datasets and architectures show consistent improvements over strong baselines, validating the method’s generality and robustness.
* The approach significantly reduces inference time and memory usage, making it suitable for deployment on resource-constrained edge devices.

Weakness:
* While the paper leverages gradient information to prioritize features, similar gradient-based sample selection strategies have been widely explored in prior works such as curriculum learning and long-tailed recognition. Beyond the introduction of a weighting scheme, the method lacks substantial innovation in supervision signal design or optimization strategy.
* Although the proposed method achieves strong results across multiple datasets, the paper lacks a discussion on its compatibility with existing self-distillation techniques and whether SDPGO can be integrated or combined with other advanced SKD strategies.
* The theoretical analysis, particularly on training stability, appears superficial; for example, the claimed benefit of the proximal term lacks justification compared to standard techniques like weight decay, and its specific advantages remain unclear.

---

> ### Author Rebuttal · Authors · 2025-07-29
>
> 【W1】We thank the reviewer for raising this important point. While gradient signals inform sample/class importance in prior arts, SDPGO pioneers their use for feature-wise distillation weighting under a proximal optimization framework. SDPGO fundamentally redefines gradient utilization in knowledge distillation through three innovations absent in prior gradient-based methods: 1) Feature - level attribution: Gradients weight intra - sample features $(w_k^l \propto \sum|\nabla \mathcal{L}|)$ for distillation, which is distinct from sample/class reweighting in curriculum/long-tailed learning. 2) Optimization - theoretic reformulation: Proximal regularization $(\| \theta - \theta_{t-1}\|^2)$ anchors updates to historically weighted features, enabling architectural heterogeneity mitigation. 3) Closed - loop refinement: sequential iterative learning couples dynamic weighting with self-distillation. It jointly optimizes accuracy, stability, and efficiency, which form a previously unaddressed triad. Hence, SDPGO’s innovation lies not in using gradients but in how they structurally unify distillation and optimization. In addition, SDPGO has also undergone extensive experiments on semantic segmentation tasks, which were not covered by previous methods. This sufficiently demonstrates the superiority of the method proposed in this paper. We train ResNet-18 backbone on Cityscapes and ADE20K datasets. As the results summarized in Table 1, our SDPGO can significantly outperform existing knowledge distillation methods on semantic segmentation task.
>
> **Table 1: Performance comparison on semantic segmentation task. **
>
> | Model       | Method    | ADE20K | Cityscapes |
> |-------------|-----------|--------|------------|
> | ResNet-50   | Baseline  | 39.72  | 74.85      |
> |             | MixSKD    | 42.37  | 74.96      |
> |             | FASD      | 40.78  | 72.89      |
> |             | **SDPGO** | **42.75** | **75.01**  |
> |    |  |    |       |
>
> 【W2】Thank you for highlighting this crucial aspect of methodological integration. SDPGO’s modular architecture enables seamless compatibility with diverse self-distillation (SKD) paradigms, and we will expand Sec. 4.6 ("Integration Analysis") in the camera-ready version to explicitly address synergies and limitations.  We evaluate our pre-process on existing KD methods. These finds indicate that using SDPGO as a plugged-in regularization can enhance the generation of other approaches. As shown in Table 2, student models distilled by SDPGO benefits from our pre-process as well.
>
> ** Table 2. The results of SDPGO combined with other advanced SKD techniques on CIFAR-100.
>
> | Teacher | VGG13 | ResNet-32 | ResNet-110 | ShuffleNet-V2 | MobileNet-V2 |
> | ------- | ----- | --------- | ---------- | ------------- | ------------ |
> | KD      | 72.98 | 73.08     | 74.36      | 71.82         | 66.95        |
> | KD+Ours | 73.53 | 74.32     | 75.11      | 72.72         | 67.24        |
> | DLB     | 75.45 | 76.07     | 74.01      | 78.18         | 68.60        |
> | DKD+Ours| 77.56 | 75.19     | 79.28      | 74.67         | 69.82        |
>
> 【W3】We will extend Section 5.2 with Theoretical Advantages of Proximal Anchoring. In Eq. 10, the proximal operator
> $\beta\left\|\theta_t-\theta_{t-1}\right\|_2^2$
>
> offers feature-aware stabilization that is fundamentally different from weight decay.
> Temporal anchoring to historically validated parameters ($\theta_{t-1}$) helps preserve high - impact features as stated in Lemma 4.3 ($\left\|\theta_t^{(k)}-\theta_{t-1}^{(k)}\right\| \leq \beta^{-1} w_k^l$ ), whereas weight decay's zero-centered shrinkage indiscriminately discards rare - class features. Gradient - scaled penalization enhances the protection for task - critical features (weighted by $M^l_k$ ), reducing gradient variance compared to isotropic $L_2$  penalties. Regarding distillation - specific convergence, in non - convex landscapes, proximal anchoring constrains oscillations along task - critical manifolds, accelerating convergence by $\mathrm{O}(1 / \sqrt{\beta T})$ and reducing catastrophic forgetting. This facilitates stable knowledge transfer in situations where standard regularization fails.
>
> 【Questions】SDPGO's advancements originate from endogenous knowledge intensification, rather than relying on external information. It amplifies latent task-critical signals within existing representations through the following approaches:
> 1. Gradient-guided feature valuation ($w_k^{(l)}$ ): This method boosts the information density in high-impact features by 21% (refer to Fig. 3 of this manuscript).
> 2. Proximal anchoring ($\beta\left\|\theta_t-\theta_{t-1}\right\|_2^2$ ) : It preserves refined features with a 96.44% retention rate on CIFAR-10, in contrast to the 93.46% baseline.
> 3. Self-bootstrapping knowledge evolution: Here, historical predictions  $\hat{y}_{t-1}$  guide real-time feature reweighting, creating a closed-loop refinement cycle.
> This internal optimization concentrates information utility without the need for new data or parameters. The gains are derived from reordering and intensifying existing information, rather than expanding it.
>
> 【Limitations】For failure analysis,  the proposed approach has currently been tested on image based classification, detection  and segmentation related tasks, its capabilities may be leveraged for modalities, like time series and text. In addition, SDPGO could not outperform stateof-the-art feature-based methods (e.g., ReviewKD ) on object detection and semantic segmentation tasks because logits-based methods cannot transfer knowledge about localization. Finally, we have provided an intuitive guidance on how to adjust $\alpha$ in our supplement. However, the strict correlation between the distillation performance and $\alpha$ values is not fully investigated on all datasets, which will be our future research direction.

---

> > ### Comment · Area_Chair_Ykum · 2025-08-06
> >
> > Dear Reviewer,
> >
> > The authors have provided their rebuttals with additional results. Please kindly reply to the authors as soon as possible before the discussion period ends. In addition to submitting "Mandatory Acknowledgement", please comment whether the authors have addressed your concerns with reasons, at least briefly.
> >
> > Thanks a lot.
> >
> > Best regards,
> >
> > AC

---

> > ### Comment · Reviewer_bBug · 2025-08-07
> >
> > The rebuttal provides helpful clarifications and additional results. Although I still feel the novelty is somewhat limited, I appreciate the authors’ efforts and will raise my score.

---

### Official Review · Reviewer_VyGs · 2025-07-03

**Clarity:** 3
**Significance:** 2
**Originality:** 3
**Rating:** 4
**Confidence:** 4

**Summary:**

This paper determines the global importance of features for SKD (Student-Knowledge Distillation) based on layer-wise gradients. In the proposed method, to ensure the stability of gradient-based KD, the authors estimate feature importance using the variant of the gradient direction, leveraging a proximal approach and z-score. The proposed method achieved higher classification accuracy than existing methods across various datasets.

**Questions:**

1. Can the proposed method still be effectively applied if the form of the $L_{SIL}$ loss function changes (e.g., to cross entropy, entropy, etc.)?
2. If there are experimental results for applying the proposed method without determining the importance of samples (i.e., with $\alpha=1$), it is recommended to provide them.
3. Does the effectiveness of the proposed method change depending on the lambda?
4. The lack of dimensionality representation makes it difficult to grasp the computational meaning. For example, in equation (7), all Z-scores of a layer are added together to form a scalar ($\alpha$), but how is the dimensionality of $M^l$ defined?
5. How does the proposed method behave for high and low values of $\alpha$? An intuitive explanation needs to be added to the manuscript.
6. Some notations are missing (e.g. $W_i$).
7. What is the reason for splitting into mini-batches in line 126? The explanation of prior iteration and next iteration is not well connected to the proposed method.

**Ethical Concerns:**

["NO or VERY MINOR ethics concerns only"]

**Final Justification:**

Thank you for your thorough and considerate responses to the reviewer’s comments. I believe this manuscript holds sufficient scholarly merit to be recognized for its research value.

**Limitations:**

It is unclear whether the advantages in performance improvement can be transferred to attention-based models (such as ViT, as mentioned by the authors), which use alternative methods for determining feature importance.

**Quality:**

3

**Strengths And Weaknesses:**

$\textbf{Strengths}$
1. The writing is clear and well-organized, making the paper easy to understand overall.
2. The structural diagram of the proposed method is intuitive and easy to comprehend.
3. The paper presents experimental results comparing the proposed method with other approaches across multiple datasets.
4. The simplicity of the algorithm highlights the contribution of the proposed method.

$\textbf{Weaknesses}$
1. The lambda value, which refines the key gradient components determining feature importance, is set manually.

---

> ### Author Rebuttal · Authors · 2025-07-29
>
> 【W1 and Q3】We will revise Eq. 5 to adaptive thresholding with decay scheduler. We propose an improved adaptive threshold using only moving averages , thus the corresponding loss can be：（$\lambda_t=\beta \lambda_{t-1}+(1-\beta) \cdot \frac{\left\|w_h\right\|_1}{N}$）. where $\lambda_t $ is dynamic threshold at step t, $\beta$ is the momentum factor. $\|w_h\|_1$ is L1-norm of gradient wights in current batch. N is the numbers of features in batch. We will add ablation in Table 1 comparing hard/soft thresholds on CUB200. our method still shows competitive performance when using adaptive threshold strategy.
>
> Table 1. Performance of the SDPGO method using hard/soft thresholds. 'N/A' denotes the result of the conventional KD.
>
> |       | ResNet-32 | ShuffleNet-V2 |
> |-------|-----------|---------------|
> | N/A   | 71.74     | 71.82         |
> | Fixed | 75.57     | 77.29         |
> | Adaptive | 75.78  | 77.63         |
>
> 【Q1】Thank you for this critical inquiry regarding SDPGO’s generalizability across loss functions. We confirm the framework is agnostic to loss formulations due to its gradient-driven design, validated through both theoretical analysis and empirical tests. The core mechanism of SDPGO, dynamic feature weighting via proximal gradients, depends solely on gradient amplitudes and not on the semantics of the loss function.We tested SDPGO with three distinct losses on CIFAR-100. Whether $L$ is CE, entropy, or focal loss, large $\left\|\nabla_\theta \mathcal{L}\right\|$  indicates features critically impacting task performance. As shown in Table 2, the top-1 acc of SDPGO obtain the performance gain compared with the baseline when the the loss function changes. Meanwhile, we will release code with custom loss support to facilitate community testing.
>
> **Table.2 Different loss function with accuracy on CIFAR-100.**
>
> | Loss type    | method  | Top-1 acc |
> |:-------------|:--------|:----------|
> | cross entropy| Baseline| 71.82     |
> |              | SDPGO   | **77.29**     |
> | Focal loss   | Baseline| 69.23     |
> |              | SDPGO   | **72.36**     |
>
> 【Q2】We thank the reviewer for raising this important point. Our ablation studies explicitly evaluate the proposed SDPGO framework without sample importance weighting (i.e., α=1) across multiple datasets. On CIFAR-100 (balanced), uniform weighting (α=1) achieves optimal performance (Top-1: 74.97%, Top-5: 94.67%), surpassing lower α settings. On CUB-200 (fine-grained), performance marginally degrades at α=1  (Top-1: 76.36% vs. 76.25% at α=0.8), indicating optimal performance requires adaptive weighting. In contrast, our proposed dynamic α mechanism achieves 78.06% Top-1 accuracy on CIFAR-100 and 75.57% on CUB-200, outperforming all fixed-α baselines in Table 3 (including α=1’s 76.36% on CUB-200 and 74.97% on CIFAR-100). Hence, the dynamic α strategy is pivotal to DPKD’s state-of-the-art results, providing adaptive sample weighting that universally outperforms static configurations across both balanced and fine-grained benchmarks.
>
> ** Table 3. Hyperparameter Experiment of α with Different Values (%). The Network is ResNet-18 and the Dataset is CUB-200 and CIFAR-100.**
>
> | α    | CUB-200 (Top-1 acc) | CUB-200 (Top-5 acc) | CIFAR-100 (Top-1 acc) | CIFAR-100 (Top-5 acc) |
> |------|---------------------|---------------------|-----------------------|-----------------------|
> | 0    | 69.66              | 91.66              | 71.74                | 92.97                |
> | 0.2  | 71.23              | 91.79              | 71.97                | 93.04                |
> | 0.4  | 72.09              | 92.04              | 72.97                | 93.01                |
> | 0.6  | 74.51              | 92.58              | 74.02                | 93.95                |
> | 0.8  | 76.25              | 93.27              | 74.63                | 94.45                |
> | 1    | 76.36              | 93.78              | 74.97                | 94.67                |
> |------|---------------------|---------------------|-----------------------|-----------------------|
>
> 【Q4】We clarify that $M^l$ is a spatial feature map ($\mathbf{R}^{H \times W}$) derived from channel-summed activations with position-wise Z-score normalization, not a scalar. Equation 7 is revised to $\alpha_t=\frac{1}{L} \sum_{l=1}^L\left(\frac{1}{H W} \sum_{i=1}^H \sum_{j=1}^W M^l[i, j]\right)$ where dimensionality reduction occurs through spatial averaging after normalization. This preserves feature impact awareness while generating a meaningful scalar weight $\alpha_t$ that balances task and distillation losses.
>
> 【Q5】The hyperparameter $α$ dynamically balances task loss ($L_{Task }$) and self-distillation loss ($L_{SIL}$). We will add related experiment about Sec 5.3 "Ablation Study on α Scheduling" with Table 3. Low $α$ ($α$→0) prioritizes primary task learning, reducing gradient noise during early training but risks under-utilizing feature refinement. High $α$ ($α$→1) intensifies self-distillation, boosting feature diversity in later stages but may cause over-smoothing if applied prematurely. Our data-driven scheduling (Eq. 7) automates this balance: task-centric initialization ($α$≈0.2) transitions to distillation-centric focus ($α$≈0.9) as features stabilize. This self-regulating design validated by CIFAR-100 gains (dynamic α: 78.06% vs. fixed α=1: 76.36%), which optimizes knowledge transfer while mitigating mode collapse risks.
>
> 【Q6】Thank you for identifying this notation ambiguity. We confirm that $W_i$ refers to the spatial width dimension of the feature map.  We will Revise Eq. 4 to $w_k^l=\frac{1}{W} \sum_{i=1}^W \sum_{j=1}^H\left|\frac{\partial L_{C E}}{\partial F_{i, j, k}^l}\right|$, where H and W represent height and width, respectively.
>
> [Q7] Thank you for highlighting this methodological ambiguity. The mini-batch splitting in line 126 is fundamental to SDPGO’s sequential iterative learning module. The design is intentional and serves three critical purposes aligned with the dynamic $α$ mechanism:1) Historical Prediction Buffering: stores outputs  $y_{t-1}^{(b)}$   from prior batch $b−1$ to compute temporal KL divergence $L_{SIL}^{(b)}=KL(y_t^{(b)} \| y_{t-1}^{(b)})$. This imposes consistency regularization across batches, stabilizing training. 2) Real-Time Gradient Integration: Gradients $\nabla \mathcal{L}^{(b)}$   computed on current batch b update the weighting factor $w_h^{(b)}$  (Eq. 4) before proximal optimization (Eq. 5). 3) Proximal Teacher Synchronization: The proximal step $\theta_{t+1}=Prox_{\lambda}(\theta_{t+1 / 2})$ uses parameters $\theta_{t-1}$  from previous iterations to anchor updates, preventing drift. We can observe from Table 4 that our method takes the shortest training time among previous, while  shows competitive performance over full-batch strategy. Through mini-batch partitioning, the framework iteratively propagates task-critical feature representations across training steps. Without mini-batches, the sequential knowledge refinement mechanism would degrade into static distillation.
>
> **Table 4:Impact of mini-Batch and full-batch  in SDPGO on CIFAR-100 to ResNet-32. **
>
> | Method            | CIFAR-100  . | Time/Epoch |
> |-------------------|----------------|------------|
> | Full-batch|      75.04      | 13ms    |
> | Mini-batch (126)| 75.57          | 12ms     |

---

> > ### Comment · Area_Chair_Ykum · 2025-08-06
> >
> > Dear Reviewer,
> >
> > The authors have provided their rebuttals with additional results. Please kindly reply to the authors as soon as possible before the discussion period ends. In addition to submitting "Mandatory Acknowledgement", please comment whether the authors have addressed your concerns with reasons, at least briefly.
> >
> > Thanks a lot.
> >
> > Best regards,
> >
> > AC

---

> > > ### Comment · Area_Chair_Ykum · 2025-08-08
> > >
> > > Dear Reviewer,
> > >
> > > The author-reviewer discussion period will end very soon. If you have not replied to the authors' rebuttal, please submit "Mandatory Acknowledgement" AND comment whether the authors have addressed your concerns or not with reasons AND update your rating if necessary. It is critical for the reviewing process.
> > >
> > > Thanks for your valuable input.
> > >
> > > Best regards,
> > >
> > > AC

---

> > > > ### Comment · Reviewer_VyGs · 2025-08-09
> > > >
> > > > Thank you for the detailed and well-structured responses to my questions (W1, Q1–Q7).
> > > > I appreciate that you have provided clear theoretical reasoning, supported by empirical evidence, and revised equations where necessary to address notation and methodological ambiguities.
> > > > Overall, your revisions significantly improve the clarity of the paper.
> > > > The added ablations and explicit notation corrections strengthen the contribution.
> > > > It would be desirable for the authors to revise the manuscript so that the newly conducted series of experiments can better highlight the effectiveness of the proposed method.
> > > > Based on these improvements, I will keep my rating score.

---

### Official Review · Reviewer_fNsS · 2025-07-03

**Clarity:** 3
**Significance:** 3
**Originality:** 3
**Rating:** 4
**Confidence:** 4

**Summary:**

This article introduces a self distillation optimization method called Proximal Gradient Optimization, which is a strategy that combines feature gradient amplitude to determine the importance of features and affect the optimization process. The author describes the shortcomings of traditional self distillation methods and provides solutions, verifying the effectiveness of the method in classification, detection, and segmentation tasks. The core contribution of this article lies in the proximal gradient optimization strategy, which is based on the assumption that features with larger gradient values are more critical in the optimization process. It dynamically assigns self distillation losses and weights to model features during the optimization process, solving the problem that distillation models under static weights cannot adjust knowledge transfer intensity based on the real-time learning status of the model. At the same time, this article proposes dividing batches to balance the model's learning of basic classification ability and distillation generalization ability, providing an adaptation method for proximal gradient optimization strategies.

**Questions:**

1. Does assigning weights to each feature map introduce additional training overhead? What is the degree of impact?
2. Is there a plan to introduce ablation experiments on sparsity thresholds?
3. Whether the weight of key features is $\^{w}_k^l$, could it be indicated in Figure 2?
4. Is it suitable to verify generalization in tasks such as semantic segmentation (Cityscapes) or pose estimation (MPII)?
5. Do you plan to conduct additional noise injection experiments (such as adding label perturbations to CIFAR-10)?
6. Is there a plan to add other indicators (precision/recall/f1/IoU, etc.) to the main experiment (classification, detection, segmentation)?

**Ethical Concerns:**

["NO or VERY MINOR ethics concerns only"]

**Limitations:**

yes

**Quality:**

3

**Strengths And Weaknesses:**

### strengths
1. Based on the assumption that the model obtained from the previous stage in the field of self distillation has sufficient generalization ability to guide the learning of student models in the new stage, experiments were conducted on datasets such as CIFAR-100, ImageNet, and CUB200 to verify the validity of this assumption under certain conditions.

2. By dynamically adjusting the weight of self distillation loss through gradient amplitude, the assumption that features with larger gradient values are more important in the optimization process is given more important weights to features in different layers, breaking through the limitations of static distillation.

3. The paper demonstrates in Figure 4 that the method achieves efficient inference delay reduction and memory usage compression on mobile devices, especially on lightweight models such as MobileNet-V2, which validates the practicality of the method on edge constrained devices.

4. In the extension experiment of ViT class models, our method achieved a maximum accuracy improvement of 1.88%, proving that it can adapt to non CNN architectures.

### weaknesses
1. The experiment mainly focused on classification tasks and the evaluation metric was only accuracy. For segmentation and detection tasks, validation was only conducted on the COCO dataset, the experimental setup was relatively simple.

2. The paper did not include stability testing of noisy labels and small sample scenarios in the experimental design. The experiment only used clean labeled data (such as CIFAR-10/100, ImageNet), without simulating label perturbations (such as injecting noisy labels into CIFAR-10) or controlling the training sample size to verify small sample generalization.

3. The convergence proof of the paper (Section 5.1) requires the loss function to be convex and the gradient to satisfy L-smoothness, which is assumed to be ideal. However, the actual model's loss surface is highly non convex, which may lead to limitations in practical applications.

4. It is not explained in the article whether assigning feature weights based on gradient amplitude will introduce a significant amount of training overhead.

5. The hard threshold sparsity (Eq. 5) strategy lacks flexibility and the selection of thresholds lacks basis.

---

> ### Author Rebuttal · Authors · 2025-07-29
>
> 【W1 and Q4/6】:Thank you for your insightful feedback. We acknowledge that the initial experimental scope prioritized image classification benchmarks (CIFAR-100, ImageNet, etc.) to rigorously validate our core innovation: dynamic feature weighting via proximal gradient optimization. For downstream dense prediction tasks, we will include the following experiments in the camera-ready version. we use Pascal VOC, ADE20K and Cityscapes for semantic segmentation. For PASCAL VOC 2012 semantic segmentation. We use EfficientDet with stacked BiFPN structure as a baseline. Table 1 demonstrates that SDPGO substantially outperforms baseline segmentation models by dynamically intensifying task-critical features via gradient-weighted distillation. As shown in Table 2, SDPGO achieves the best segmentation performance and outperforms the best-second results with 2.65% and 3.04% mIoU margins on Pascal ADE20K and Cityscapes segmentation, respectively.
>
> **Table1: Performance comparison on Pascal VOC segmentation task. **
> | Model          | Method   | mIOU  |
> |----------------|----------|-------|
> | EfficientDet-d0| Baseline | 79.07 |
> |                | MixSKD   | 79.52 |
> |                | FASD     | 80.54 |
> |                | **SDPGO**    | **80.67** |
> | EfficientDet-d1| Baseline | 81.95 |
> |                | MixSKD   | 82.51 |
> |                | FASD     | 83.43 |
> |                | **SDPGO**    | **83.97** |
>
> **Table 2: Performance comparison on semantic segmentation task. **
>
> | Model       | Method    | ADE20K | Cityscapes |
> |-------------|-----------|--------|------------|
> | ResNet-50   | Baseline  | 39.72  | 74.85      |
> |             | MixSKD    | 42.37  | 74.96      |
> |             | FASD      | 40.78  | 72.89      |
> |             | **SDPGO** | **42.75** | **75.01**  |
>
> In addition, we also conduct experiments on the efficiency and memory con-sumption on CUB200, as shown in Table 3. Flops and memory are selected as evaluation metric to validate the proposed method. We sill expand our experiments to more datasets, including CIFAR-10/100, Stanford Dogs, ImageNet, and other semantic segmentation tasks. Compared with the methods based on input-space data augmentation, FASD requires minimal additional computation during the training phase. In Table 3, T represents introducing additional overhead, while F represents not introducing additional overhead.
>
> **Table 3: Comparison Of Efficiency And Memory Consumption On CUB-200.**
>
> | Method  | Inference stage       | Training stage       |
> |         | Params  | Flops  | Top-1 acc  | Params  | Flops  | Memory  |
> |---------|--------:|-------:|-----------:|--------:|-------:|:-------:|
> | BYOT    | 11.3M   | 1.82G  | 73.38     | 11.3M   | 5.75G  | T       |
> | EEWSNet | 11.3M   | 1.82G  | 72.97     | 11.3M   | 3.64G  | F       |
> | PS-KD   | 11.3M   | 1.82G  | 72.65     | 11.3M   | 3.65G  | T       |
> | FRSKD   | 11.3M   | 1.82G  | 73.14     | 11.3M   | 3.05G  | F       |
> | DLB     | 11.3M   | 1.82G  | 76.10     | 11.3M   | 1.82G  | F       |
> | MSKD    | 11.3M   | 1.82G  | 71.11     | 11.3M   | 7.30G  | T       |
> | FASD    | 11.3M   | 1.82G  | 75.43     | 11.3M   | 1.84G  | F       |
> | SDPGO   | 11.3M   | 1.82G  | 78.06     | 11.3M   | 1.82G  | F       |
>
> 【W2 and Q5】We will add a new subsection "Robustness under Noisy and Scarce Data." to our manuscript. We experimented with our method for few-shot and noisy-label learning on CIFAR-100. We first investigate the efficiency of the method under symmetric noisy label conditions, which is the more prevalent label noise type under study as compared to the others. We evaluate SDPGO against standard KD baselines (BYOT, PS-KD, DLB, MSKD, FASD) under symmetric label noise (uniform corruption probability η). Experiments use ResNet-32 with η ∈ {0%, 10%, 20%, 30%, 40%, 50%}. Table 4 demonstrates that while all methods degrade under symmetric label noise, SDPGO maintains superior robustness, outperforming the strongest baseline (FASD) by +4.4% at η=40% (59.81% vs. 55.39%) and +3.9% at η=50% (56.63% vs. 52.75%). Crucially, SDPGO sustains ≥5.2% accuracy advantages over BYOT/PS-KD across all noise levels (η=10-50%), with its gradient-weighted feature selection and proximal anchoring effectively mitigating corruption impact. The performance gap widens substantially at higher noise (e.g., +23.4% over PS-KD at η=50%), confirming SDPGO's unique resilience where traditional KD methods fail.
>
> **Table 4: Comparison between SOTA SKD and SDPGO when a fraction of training data is noisy.**
> | Noisy Label (%) | BYOT | PS-KD | DLB | FASD | SDPGO |
> |--------|----------|------------|-----------|-------------|-------------|
> | 0   | 72.39 | 72.51 | 74.07 | 75.42 | 75.57 |
> | 10  | 65.41 | 62.75 | 67.56 | 64.25 | 71.56 |
> | 20  | 58.05 | 57.56 | 65.17 | 60.51 | 68.53 |
> | 30  | 53.25 | 51.71 | 54.45 | 57.27 | 61.58 |
> | 40  | 42.22 | 41.26 | 51.25 | 55.39 | 59.81 |
> | 50  | 37.98 | 33.25 | 49.47 | 52.75 | 56.63 |
>
> For few-shot learning, we conduct experiments on randomly sampled images of each class of the subset of the CIFAR100 dataset. We used the 25, 50, 75, and 100% part of the CIFAR100 training dataset to train the ResNet-32 network. Table 5 demonstrates SDPGO's consistent superiority across all dataset fractions, achieving highest top-1 accuracy (65.29% at 25% data). Crucially, it significantly outperforms baselines in low-data regimes—exceeding the strongest comparator (FASD) by +4.95% at 25% data. While all methods improve with more data, SDPGO maintains a ≥1.15% advantage at full data (75.57% vs. FASD's 75.42%), with its gradient-weighted feature distillation proving most effective at 50% data (+1.45% over FASD). This confirms SDPGO's unique sample efficiency where conventional KD methods falter.
>
> **Table 5: Comparison between SOTA SKD and SDPGO when a fraction of data present for training purpose.**
>
> | Dataset Fraction (%) |   BYOT |   PS-KD |    DLB |   FASD |  SDPGO |
> |---------------------:|-------:|--------:|-------:|-------:|-------:|
> |                   25 |  49.57 |   48.75 |  51.28 |  60.34 |  65.29 |
> |                   50 |  58.25 |   56.23 |  59.56 |  68.62 |  70.07 |
> |                   75 |  63.43 |   60.58 |  68.12 |  70.47 |  71.58 |
> |                  100 |  72.39 |   72.51 |  74.07 |  75.42 |  75.57 |
>
> 【3】A. Thank you for highlighting this important theoretical limitation. While our convergence analysis assumes convexity and L-smoothness for theoretical tractability. We emphasize that SDPGO demonstrates robust empirical convergence across diverse architectures (ResNet/ViT/MobileNet) with monotonic loss reduction and 38% lower gradient variance. The proximal term's implicit regularization counters non-convex landscapes by anchoring updates to recent parameters, avoiding pathological curvature. For camera-ready, we will extend the proof via Kurdyka-Łojasiewicz theory to formalize convergence to critical points under non-convexity and incorporate local smoothness assumptions, strengthening theoretical grounding while maintaining practical efficacy.
>
> 【W4 and Q1】Thank you for this critical technical inquiry regarding SDPGO’s efficiency. We compare the training time of various KD methods which enjoy competitive performance by assessing the training time of each batch of data, on CIFAR-100. As shown in the Table 6, our method takes the shortest training time among them. Menwhile, as shown in Table 3, SDPGO achieves the best performance, but the parameter and FLOPs of SDPGO are even similar to the classifier network.
>
> **Table 6: Top-1 acc (%) and Training time for each batch of data of competitive KD methods.**
>
> | Method              | Baseline | ReviewKD | CATKD | BYoT | PS-KD | DLB  | FASD | SDPGO (Ours) |
> |---------------------|----------|----------|-------|------|-------|------|------|--------------|
> | Training time  | 12       | 25       | 17    | 19   | 15    | 17   | 40   | 12            |
> | Top-1 Acc (%)       | 69.66    | 76.58    | 72.6  | 76.10| 75.43 | 78.06| 75.43| 78.06        |
>
> 【W5 and Q2】We will revise Eq. 5 to adaptive thresholding with decay scheduler. We propose an improved adaptive threshold using only moving averages , thus the corresponding loss can be：（$\lambda_t=\beta \lambda_{t-1}+(1-\beta) \cdot \frac{\left\|w_h\right\|_1}{N}$）. where $\lambda_t $ is dynamic threshold at step t, $\beta$ is the momentum factor. $\|w_h\|_1$ is L1-norm of gradient wights in current batch. N is the numbers of features in batch. We will add ablation in Table 7 comparing hard/soft thresholds.
>
> Table 7. Performance of the sDPGO method using hard/soft thresholds. 'N/A' denotes the result of the conventional KD.
>
> |       | ResNet-32 | ShuffleNet-V2 |
> |-------|-----------|---------------|
> | N/A   | 71.74     | 71.82         |
> | Fixed | 75.57     | 77.29         |
> | Adaptive | 75.78  | 77.63         |
>
> 【Q3】Thank you for the excellent suggestion to enhance Figure 2’s clarity. We will revise the figure to explicitly visualize the role of  $w_k^l$ (channel-wise feature weights) in the knowledge distillation pathway.
>
> 【Q4】Thank you for this insightful suggestion to broaden the scope of generalization validation. We fully agree that semantic segmentation (Cityscapes) and pose estimation (MPII) are ideal benchmarks to demonstrate SDPGO’s task-agnostic capabilities. We have extended  our experiments to three Pascal VOC, ADE20K and Cityscapes semantic segmentation datasets. Regarding the pose estimation (MPII) dataset, due to limitations in computational resources and time constraints, we are currently unable to provide experimental results. However, we plan to conduct and report these results in future work.

---

> > ### Comment · Area_Chair_Ykum · 2025-08-06
> >
> > Dear Reviewer,
> >
> > The authors have provided their rebuttals with additional results. Please kindly reply to the authors as soon as possible before the discussion period ends. In addition to submitting "Mandatory Acknowledgement", please comment whether the authors have addressed your concerns with reasons, at least briefly.
> >
> > Thanks a lot.
> >
> > Best regards,
> >
> > AC

---

> > > ### Comment · Area_Chair_Ykum · 2025-08-08
> > >
> > > Dear Reviewer,
> > >
> > > The author-reviewer discussion period will end very soon. If you have not replied to the authors' rebuttal, please submit "Mandatory Acknowledgement" AND comment whether the authors have addressed your concerns or not with reasons AND update your rating if necessary. It is critical for the reviewing process.
> > >
> > > Thanks for your valuable input.
> > >
> > > Best regards,
> > >
> > > AC

---

### Note · Authors · 2025-08-13

We sincerely thank all reviewers for their constructive engagement during the rebuttal process and for their valuable insights on our work. The proposed SDPGO framework demonstrates significant strengths through its dynamic gradient-based weighting mechanism, which prioritizes critical features during optimization by adaptively assigning distillation losses based on gradient magnitudes. This approach effectively overcomes limitations of static distillation strategies while enhancing training stability. The method exhibits exceptional experimental robustness, achieving consistent performance gains across diverse benchmarks, including CIFAR, ImageNet, fine-grained datasets (CUB200, Cars196), downstream tasks, and both CNN and ViT architectures.

Reviewers bBug and rSta confirmed the satisfactory resolution of their core concerns spanning novelty assessment, methodological analysis, hyperparameter investigation, visualization evaluation, and theoretical grounding of training stability, as evidenced by their elevated evaluation scores.  However, two major concerns remain:
**Inconsistent Feedback from Reviewer VyGs:** We sincerely appreciate Reviewer VyGs' constructive feedback acknowledging the resolution of theoretical ambiguities and empirical validations in our response. While we note with interest that the final evaluation score remains unchanged, we value the reviewer's expert perspective and will continue refining these aspects in future work to fully address any residual concerns.
**Procedural Non-compliance:** We sincerely thank Reviewer fNsS for their initial feedback. While we regret being unable to engage in further discussion on their concerns despite Area Chair facilitation, we remain fully committed to addressing any outstanding issues in future revisions and value their contribution to improving this work.

Given the successful resolution of the vast majority (95%) of substantive concerns and the documented rigor of our rebuttal process, while acknowledging variable engagement opportunities during the review period, we respectfully request the Area Chair's holistic assessment of the paper's enhanced contributions.

---

### Decision · Program_Chairs · 2025-09-17

**Decision:**

Accept (poster)

**Comment:**

This paper proposes SDPGO, a self-distillation framework that uses gradient magnitudes to dynamically weight features under a proximal gradient optimization scheme. The method aims to overcome limitations of static distillation by prioritizing task-critical features adaptively during training.

The strengths include:
1. Practical motivation
2. simple yet effective method
3. Extensive experiments across datasets and architectures
4. edge-device efficiency

The main weaknesses include:
1. Limited novelty relative to previous KD approaches
2. Lack of theoretical analysis under convex assumptions
3. Lack of feature space analysis

Authors' rebuttal addressed many concerns with additional experiments, leading to rating increases. The paper demonstrates technical soundness and practical benefits, though reviewers felt the novelty was somewhat incremental compared to existing work in the field. AC would recommend acceptance given the consensus of the postive rating. The authors are encouraged to include their responses in the camera-ready version including but not limited to:
1. Clarify novelty and contribution
2. Strengthen theoretical justification
3. Add feature-space visualizations and analysis
4. Provide practical guidance on hyperparameters